# Asymmetric and Symmetric Dimethylarginines as Renal Function Parameters in Paediatric Kidney Diseases: A Literature Review from 2003 to 2022

**DOI:** 10.3390/children9111668

**Published:** 2022-10-31

**Authors:** Michalina Jezierska, Joanna Stefanowicz

**Affiliations:** 1Department of Paediatrics, Haematology and Oncology, Faculty of Medicine, Medical University of Gdansk, 7 Debinki Street, 80-211 Gdansk, Poland; 2Department of Paediatrics, Haematology and Oncology University Clinical Centre, 7 Debinki Street, 80-952 Gdansk, Poland; 3Faculty of Health Sciences, Medical University of Gdansk, 7 Debinki Street, 80-211 Gdansk, Poland

**Keywords:** dimethylarginine, asymmetric dimethylarginine, symmetric dimethylarginine, children, kidney, paediatric kidney disease

## Abstract

Asymmetric dimethylarginine (ADMA) and symmetric dimethylarginine (SDMA), inhibitors of nitric oxide synthase, play important roles in many processes in the body. Most data in the literature concern their importance in adult chronic kidney disease (CKD). According to them, SDMA well reflects the glomerular filtration rate (GFR), and higher ADMA concentrations are associated with hypertension and higher mortality. In addition, both substances are recognised cardiovascular risk factors in CKD. The purpose of this review was to summarise the studies on dimethylarginines in renal diseases in children, about which we have much fewer data. The review focuses specifically on dimethylarginine’s relation to routinely used renal function parameters. Finally, we analysed 21 of the 55 articles published between 2003 and 2022 on dimethylarginines in kidney diseases in children (from birth to 18 years of age), obtained by searching PubMed/MEDLINE (search terms: “dimethylarginine” and “kidney”).

## 1. Introduction

The history of dimethylarginines in medicine began in 1970, when they were first isolated from human urine [1]. At present, asymmetric dimethylarginine (ADMA) and symmetric dimethylarginine (SDMA) are recognised uraemic toxins [2]. We know that they play a role in many human diseases. Their wide participation in physiological and pathological processes in the organism is because they act mainly as endogenous inhibitors of nitric oxide (NO) synthesis [3,4,5,6]. Actual knowledge about dimethylarginines is based primarily on research involving adults. Studies on ADMA and SDMA in the paediatric population represent a small percentage of them. This review focuses on the assessment of dimethylarginines as renal function parameters in paediatric kidney diseases. In addition, brief information on the synthesis, metabolism, and pathophysiology of dimethylarginines is presented.

For the purpose of this review, the literature on ADMA and SDMA in children (from birth to 18 years of age) with kidney diseases from the PubMed/MEDLINE database published between 2003 and 2022 was analysed. The used search terms were “dimethylarginine” and “kidney”. The initial search yielded 55 papers, of which the final analysis was 21. Excluded from the analysis were 16 studies with adults as the study population, 10 paediatric studies unrelated to kidney disease, 5 papers that addressed laboratory diagnostic methods, 2 review papers, and 1 letter to the editor. Most of the articles were about chronic kidney disease (CKD). The others were about solitary functioning kidney (SFK), nephrotic syndrome (NS), sporadic focal segmental glomerulosclerosis (FSGS), and haemolytic–uraemic syndrome (HUS).

## 2. Dimethylarginines Synthesis, Metabolism, and Pathophysiology

Dimethylarginines are formed by posttranslational protein methylation of arginine residues. Type I and type II protein arginine methyltransferases (PRMTs) are responsible for this process. The donor of the methyl group in this reaction is S-adenosylmethionine. The attachment of two methyl groups to one of the terminal nitrogen atoms of the guanidino group of arginine by PRMT I results in the formation of NG, NG-dimethylarginine (ADMA). Its optical isomer, SDMA, is formed in a reaction by PRMT II [7].

ADMA and SDMA are released into the extracellular space and circulation by proteolysis. They are small molecules and are easily removed by dialysis. Their transport into and out of cells, as with arginine, is mediated by the cationic amino acid transporter (CAT) [8]. ADMA is metabolized mainly by dimethylarginine dimethylaminohydrolase-1 and dimethylarginine dimethylaminohydrolase-2 (DDAH-1, DDAH-2, respectively) to citrulline and dimethylamine (DMA) and by alanine-glyoxylate aminotransferase 2 (AGXT2) [9,10]. Only approximately 10–20% of ADMA is excreted unchanged by the kidney. In the case of SDMA, it is the opposite; it is primarily removed by the kidneys, with little metabolism by the abovementioned enzymes [11]. The pathways of dimethylarginines metabolism explain what factors determine the concentrations of both substances. For ADMA the most important is the activity of DDAHs and AGXT2, and for SDMA it is the filtration function of the kidney.

Dimethylarginines inhibit NO production and decrease its bioavailability, thus affecting all body processes in which it participates. NO is synthesised by nitric oxide synthase (NOS), and the substrate for this process is L-arginine. ADMA acts as a competitive inhibitor of NOS. Therefore, the Arg/ADMA ratio is considered an exponent of NO bioavailability [11]. In addition, ADMA reduces NOS activity by inhibiting its phosphorylation [12]. SDMA inhibits NO production indirectly by inhibiting arginine uptake by CAT [13]. Furthermore, in rat studies, it also inhibited L-arg tubular reabsorption in the loop of Henle [14]. Both mechanisms of action of SDMA contribute to the reduction in substrate concentrations for NO production.

## 3. Dimethylarginines Measurements

Dimethylarginines concentrations can be measured in plasma, urine, and tissues. According to the literature, these concentrations do not always correlate with each other [15]. The tissue dimethylarginine concentrations are higher than those in plasma [8]. For measurements, we can use high-performance liquid chromatography (HPLC), mass spectrometric (MS)-based methods, or enzyme-linked immunosorbent assay (ELISA) [16,17,18,19,20,21,22]. The values will differ among these methods. The most accurate results are obtained using MS-based methods. The ELISA method gives overestimated results [23,24,25]. 

## 4. Dimethylarginines as Renal Function Parameters

### 4.1. Healthy Children

Dimethylarginines participating in the regulation of NO bioavailability are one of the elements involved in maintaining body homeostasis. An attempt to assess dimethylarginines as parameters of renal function in children under homeostatic conditions was made by Jaźwińska-Kozuba et al. [26] The study was conducted on a group of 40 healthy children. In the study, ADMA and SDMA were marked by liquid chromatography–tandem mass spectrometry, creatinine was measured by the Jaffe assay, and eGFR was calculated based on the Schwartz formula. The SDMA concentration and SDMA/ADMA ratio correlated positively with creatinine and negatively with eGFR, and the SDMA/ADMA ratio correlation was stronger than that of the SDMA. No correlation was found for ADMA. This is the only report on this subject in healthy children that we found. Nevertheless, it is very valuable because it can be a point of reference for evaluating the same parameters in the paediatric population under conditions of impaired homeostasis, for example, in the case of kidney diseases.

### 4.2. Chronic Kidney Disease (CKD) and Cardiovascular Disease (CVD)

When homeostasis is disturbed by abnormalities of kidney function and structure presented for >3 months with implications for health, we recognise chronic kidney disease (CKD) [27]. The pathomechanism of CKD is complex and not yet fully understood. In view of the natural course of this disease, it seems crucial to intervene clinically as soon as possible to stop or slow down the disease and prevent its complications. It has been proven that in adults with CKD, SDMA correlates with creatinine clearance and eGFR, and a higher level of ADMA is associated with hypertension and higher mortality [3,28,29]. Both dimethylarginines are the link between CKD and cardiovascular disease (CVD), the main cause of mortality in CKD, and are recognised as independent cardiovascular risk factors [29].

Wasilewska A. et al., examined 35 children with CKD stages 1–5 (children on dialysis and after transplantation were excluded) and 42 healthy children [30]. In this study, creatinine, cystatin C, serum SDMA and eGFR were compared between the groups. Both cystatin C and SDMA concentrations were significantly higher in the patients with CKD. Additional subgroup analysis of patients with CKD: A (CKD 1 and 2), B (CKD 3), and C (CKD 4 and 5) showed significant differences in SDMA concentrations between Groups A and B. There were no such results for cystatin C. Both cystatin C and SDMA correlated positively with creatinine in all patients and negatively with eGFR in CKD patients. The correlation with eGFR was stronger for SDMA. Further statistical analysis showed that SDMA was better than cystatin C at identifying CKD stages and detecting patients in Group A.

According to a study in adults, dimethylarginines concentration increases not only in CKD but also in its progression. Brooks E. R. et al., based on this information, unable tofully explain a GFR decline in CKD by traditional markers and they knowing that dimethylarginine excretion is relatively constant with normal GFR, have attempted to evaluate whether baseline serum dimethylarginine concentrations contribute to baseline eGFR and eGFR decline in paediatric patients [31]. For this purpose, they examined 352 children with CKD over a period of 4 years, via at least 2 medical appointments (baseline and visit after 2 or 4 years). According to their results, baseline dimethylarginine concentration was inversely correlated with baseline eGFR, but there was no correlation with eGFR rate of decline.

The studies cited above show that SDMA may be useful for assessing mild and medium eGFR decreases in children, but its use to assess changes in eGFR is questionable and requires further analysis. The results may have been influenced by the relatively small size of the study population, especially in the first study. Differences in CKD aetiology in children and adults may also have played a role. Children develop CKD mainly because of congenital abnormalities of the kidney and urinary tract (CAKUT). In adults, the dominant glomerular aetiology is principally caused by diabetes, hypertension, and glomerulonephritis [27]. Similar limitations apply to all studies involving children with CKD as the study population. This may explain the inconsistency of their results and conclusions with those of studies in adults.

As CKD progresses, uraemic toxins, mainly small and medium molecular products of protein metabolism, accumulate in the blood. Snauwaert E. et al. evaluated to what extent eGFR reflects uraemic toxin accumulation in children with CKD [32]. In a group of 65 children with CKD, concentrations of small solutes (including ADMA and SDMA), middle molecules, and protein-bound solutes were compared with eGFR. In this study, eGFR correlated well with SDMA and poorly with ADMA. This result is consistent with the results of studies both in adults and with a few studies of this type in children, and it is not surprising. However, truly interesting in this study is that in a simple way it demonstrates how complex the pathomechanism of CKD is and that there is no one perfect substance/marker to reflect it. For the multidimensional evaluation of patients with CKD, to diagnose the disease and its staging and to assess its complications and associated risk, it seems necessary, in addition to clinical assessment, to evaluate a panel of parameters that have proven importance in the course of CKD. The same team of researchers again led by Snauwaert E. in a 2018 paper attempted to establish reference values for selected uraemic toxins in a group of nondialysed children with CKD [33]. In a study of 57 children with CKD stages 1–5 and 50 healthy children, SDMA levels were elevated in the study group compared to the control group. The differences between groups were highest in CKD stages 1 and 2, which is consistent with the results of the abovementioned Wasilewska et al. Moreover, the difference in the normalised concentrations of SDMA was greater than that for creatinine. This opens a discussion about the greater sensitivity of SDMA than creatinine in detecting eGFR decline, which needs to be supported by further studies.

A similar study on uraemic toxins was conducted by Benito S. et al., who examined 16 substances from the arginine–creatine, urea cycle, and arginine methylation metabolic pathways in 32 children with CKD and compared these results with the results of 24 healthy controls [34]. In this study, ADMA and SDMA had higher concentrations in the study population than in the control group, but unlike the Snauwaert E. et al., study, similar effectiveness to creatinine in the early diagnosis of CKD. Additional use of multivariate analysis of plasma concentration obtained by targeted metabolomics LC-QTOF showed that using plasma concentrations of creatinine, citrulline, SDMA, and S-adenosylmethionine together increased the efficiency of diagnoses by 18% compared to creatinine-only determination and improved the classification into the other stages of CKD. Thus, the superiority of broader laboratory evaluation over single-substance determination in the evaluation of CKD was again demonstrated. Benito et al. expanded their study by developing and validating a new LC-QQ-MS analytical method in paediatric patients with CKD and published these findings in 2019 [35].

CKD and cardiovascular disease (CVD) are inextricably connected. Endothelial dysfunction is the process that links them together. Dimethylarginines have a proven role in its development, primarily as inhibitors of nitric oxide synthase but also by other mechanisms, which we describe later in this section. Although cardiovascular events in children occur less frequently than in adults, identification of risk factors and early implementation of treatment, with individualisation of management in CVD, may benefit this group in adulthood. Thus, there are increasing attempts to determine the cardiovascular risk profile in children with CKD. A study that evaluated the association of methylated L-arginine (L-arg) derivatives with eGFR and their effect on the burden of hypertension was conducted by Brooks et al. [36]. The study included 28 children with CKD stages 2–3 and a control group of 10 of their siblings. Serum L-arg, ADMA, SDMA, and creatinine measurements were performed twice, 3 months apart. Blood pressure measurement was performed with 24 h ambulatory blood pressure monitoring (ABPM). ADMA, SDMA, and SDMA/ADMA were higher, and L-arg, L-arg/ADMA, and L-arg/SDMA were lower in the study group than in the control group. eGFR correlated with SDMA, SDMA/ADMA, and L-arg/SDMA in the combined analysis of both groups. Similarly, SDMA and SDMA/ADMA correlated positively with awake (AW) and asleep (AS) systolic (S) and diastolic (D) blood pressure (BP) loads in both groups. In the CKD group, with constant eGFR, SDMA accounted for 27% of the variability in DBP load in AW. No association of BP with ADMA was demonstrated. The study results at both time points (3-month interval) were comparable. This study again confirmed the value of SDMA as an eGFR decrease marker but also identified it as a potential indicator of BP variability in 24 h measurements. Similar studies were performed by Lin I.C. et al., Chien S.J. et al., Hsu C.N. et al., and Kuo H.C. et al., but they extended their analysis to evaluate vascular parameters such as carotid intima-media thickness (cIMT), flow-mediated dilatation (FMD), pulse wave velocity (PWV), augmentation index (AI) or ABPM-derived arterial stiffness index (AASI) in different combinations [37,38,39,40]. Lin I.C. et al., and Shao-Ju Ch. et al., evaluated children with CKD stages 1–3. The first study included 55 children and the second 57 children. In the first study, negative correlations were found between AI and Arg/ADMA, AASI and SDMA, awake SBP and DMA/ADMA; positive correlations were found between DMA/ADMA and AASI, awake SBP and SBP. In the second study, the Arg/ADMA ratio correlated with systolic BP values and left ventricular mass. In addition, the ADMA, SDMA, and Arg/ADMA ratio correlated with arterial stiffness. Hsu C.N. et al., and Kuo et al., obtained their results by studying 125 and 45 children with stages 1–4 CKD, respectively. In Hsu C. N. et al.’s study, children with abnormal BP values had a lower Arg/ADMA ratio but a higher ADMA/SDMA ratio. In Kuo H.C. et al.’s study, where urinary concentrations of ADMA and SDMA were assessed, the Arg/ADMA ratio was higher, and the ADMA/SDMA ratio was lower in children with CKD 2–4 compared to CKD 1. A higher (ADMA + SDMA)/Arg ratio was correlated with abnormalities in ABPM. A very important result of all the above-mentioned studies was the finding of abnormal BP values in ABPM in approximately 2/3 of the studied children with CKD, including patients with CKD stage 1. This was also confirmed in studies where the association of dimethylarginines with eGFR and blood pressure abnormalities was not as obvious, such as the study by Lin Y.J. et al., which involved 44 children with CKD 1–3 [41]. To summarise, studies in children with CKD, similar to adult research, suggest that one of the mechanisms coresponsible for the development of hypertension is the reduced bioavailability of NO. Dimethylarginines, as the most potent endogenous NOS inhibitors, are involved in this process. The evaluation of the clinical usefulness of dimethylarginines in this regard seems to be extremely important. The same studies showed that blood pressure abnormalities are present in a high percentage of children with CKD and in the low stages of the disease. These abnormalities were revealed in only ABPM measurements. Office measurements turned out to be an imperfect tool, masking the problem. This provides another reason why the identification of additional cardiovascular risk factors can be a step towards improving diagnosis and accelerating our efforts. Most studies indicate three groups of potential parameters that may be of key importance for the assessment of cardiovascular risk in children: abnormalities in blood pressure values, concentrations of CIT-Arg-NO (citrulline-arginine-nitric oxide) pathway derivatives (including dimethylarginines) and vascular parameters, especially arterial stiffness.

As we wrote at the beginning of this subsection, the role of dimethylarginines in promoting CVD in CKD is not limited to the inhibition of NO synthesis. Speer T. et al. proved that SDMA in children and adults with CKD is responsible for HDL transformation into “toxic” HDL, which loses its endothelial protection property, and activate of a Toll-like Receptor (TLR) that promotes endothelial dysfunction [42]. The same team led by Shroff R. conducted a study on this issue to confirm the previous results and additionally evaluated plasma and direct HDL SDMA concentrations in a group of 82 children in various stages of CKD compared to 12 healthy children [43]. In the study, children with CKD had significantly higher SDMA levels than those of the control group, with the highest values observed in children on dialysis. Similar results were obtained for SDMA measurements in HDL. The results of the cited studies may indicate that SDMA plays a key role in the development of HDL dysfunction in CKD.

Dimethylarginines, as NO-related parameters and potential cardiovascular risk factors in children with CKD, are sometimes used not as the main object of study but as a reference. For example, Hsu C.N. et al. conducted a survey on the connection between acrylamide metabolites in urine and cardiovascular risk factors in 112 children with CKD 1–4 [44]. At the same time, they stated that children with CKD 2–4 had higher SDMA levels than those with CKD 1. Similarly, Drożdż D. et al. evaluated thrombomodulin as a marker of endothelial dysfunction in 59 children with CKD 1–5, and Makulska I. et al., tested skin autofluorescence (sAF), a marker of the accumulation of advanced glycation end products, as a vascular damage indicator in 76 children with CKD [45,46]. In the last study, sAF was higher in children with CKD than in controls and positively correlated with ADMA levels. As a noninvasive diagnostic method, sAF would be particularly valuable in the paediatric population.

### 4.3. Other Kidney Diseases

A natural consequence of the interest in dimethylarginines in kidney diseases was their evaluation in renal transplant patients. A study of 26 children after renal transplantation and their comparison with a control group was conducted by Andrade et al. [47]. In this study, only ADMA evaluation was used, but in both blood and urine. ADMA serum concentrations were significantly higher and the Arg/ADMA ratio was lower in the study group than in the control group. These results may indicate a disturbance in the methylation cycle in renal transplant patients that results in decreasing NO bioavailability and thus endothelial dysfunction and increasing cardiovascular risk. Given the evidence that CVD is the leading cause of death in patients with end-stage renal disease (ERSD), the results of this study encourage further exploration of the topic.

Solitary functioning kidney (SFK) is the next clinical situation where we can expect an increased risk of renal dysfunction. The KIMONO study showed that 1/3 of children with SFK had symptoms of kidney injury [48]. It is not dependent on the cause of kidney loss, which can be unilateral renal agenesis (URA) or unilateral nephrectomy (UN). The first attempt to assess the SDMA in this population was made by Tarnata-Janusz K. et al. [49]. They compared SDMA concentrations between 51 children with SFK (URA and UN groups) and 21 healthy children. SDMA levels were higher in children with SFK than in the control group, but there were no differences between the URA and UN groups. SDMA showed a positive correlation with creatinine concentration and a negative correlation with eGFR, which was in line with previous reports on SDMA. Additionally, they assessed the area under the curve (AUC) for creatinine and SDMA in study populations to assess the sensitivity and specificity of SDMA for eGFR assessment, but there were no differences. In view of the results of this study, the superiority of SDMA over creatinine for evaluating eGFR cannot be confirmed.

Filtration membrane damage in nephrotic syndrome (NS) and dyslipidaemia occurring in this disease make NS a perfect candidate to test dimethylarginines. Hyla-Klekot L. et al. tried to assess SDMA as an eGFR decrease marker and ADMA as a marker of atherosclerosis in children with NS [50]. In a group of 32 children with NS based on primary glomerulonephritis with normal eGFR, they measured L-arg, ADMA, SDMA, biochemical parameters, lipid profile, ionogram, and aminotransferases before treatment and in the remission/relapse phase of the disease. In all patients and in all phases of the disease, SDMA negatively correlated with eGFR, which confirms the usefulness of SDMA in assessing renal filtration rate. ADMA correlated negatively with cholesterol and triglycerides, traditional CVD risk factors, but only in the relapse phase, which raises questions about the usefulness of ADMA as a marker of atherogenesis in this patient group.

The cause of sporadic focal segmental glomerulosclerosis (FSGS) remains unknown. There is a hypothesis according to which circulating factor inhibits NOS, which promotes glomerulosclerosis. Lücke T. et al. assessed the L-arg/NO pathway in 9 children with FSGS, 11 children with renal diseases other than FSGS, and 9 healthy children [51]. In all groups, they measured ADMA both in plasma and urine. The ADMA serum concentration was significantly higher in patients with FSGS than in controls and and higher but without statistical significance in patients with other renal diseases than controls. There were no eGFR differences between groups, so an increased level of ADMA was not an effect of renal impairment. Urinary excretion of ADMA and DMA in patients with FSGS was higher than in healthy controls; thus, ADMA metabolism in FSGS group seemed to be normal. Additionally, an inverse correlation between ADMA and eGFR was found only in patients with FSGS. Based on these study results, we suspect that ADMA, as a NOS inhibitor, can be involved in FSGS pathogenesis, which is consistent with the original hypothesis of this study.

In only one article from the analysed literature, dimethylarginines were assessed in the setting of acute kidney injury. Kanzelmyere N. K. et al., attempted to evaluate the L-arginine/NO pathway in children with the haemolytic–uraemic syndrome (HUS), the most common cause of acute kidney injury in children [52]. Based on a study of 12 children with typical HUS and 12 healthy children, they proved that the L-arginine/NO pathway is disturbed in HUS. In the study, the plasma ADMA concentration was slightly lower, with a concomitant significantly lower renal excretion rate of ADMA in children with HUS compared to the control group. In view of the similar plasma L-arginine and urinary DMA concentrations in both study groups, it seems that the differences in ADMA concentrations may be due to a decrease in ADMA synthesis in HUS. The mechanism underlying these changes may be involved in the pathomechanism of the disease but remains unclear for the time being.

Table 1 summarises the articles used in the review by disease type, study population, analytic methods, and key findings.

## 5. Conclusions

The aim of this review was to summarise the current knowledge on dimethylarginines as a parameter of renal function in kidney diseases in the paediatric population. As shown in our analysis of the literature from 2003 to 2022, SDMA correlates with creatinine, cystatin C, and eGFR. It appears to be a good indicator of mild to moderate eGFR decline and thus may be useful in the initial diagnosis of CKD. However, its superiority over routinely used renal function parameters, as well as its role as a marker of disease progression, are uncertain. Both ADMA and SDMA appear to be involved in the pathomechanism of the development of hypertension in children with CKD. However, they currently lack the status of independent cardiovascular risk factors in the paediatric population, unlike adults. Speer et al., and Shroff et al., reported on the role of SDMA in the formation of “toxic” HDLs, shed new light on this topic, and provide a field for further research.

The studies on dimethylarginines in other kidney diseases, such as SFK, NS, FSGS, and HUS, are single papers that do not allow for the drawing of clear conclusions.

An important and undoubted limitation of paediatric studies is the small size of the study population and thus the lower power of the study compared to adult studies. The results of adult studies cannot be directly translated to children because of the differences between these two populations. However, they can influence the design and implementation of new research in the paediatric population.

## Figures and Tables

**Table 1 children-09-01668-t001:** The role of ADMA and SDMA in paediatric kidney diseases—summary of the articles and presented studies from 2003 to 2022 used in the review according to disease type, study population, used analytic methods, and key findings.

Type of Disease	Study	Study Population	Analytic Methods	Results
**CKD**	Wasilewska A. et al., 2012 [30]	35 CKD stage 1–5 cases 42 controls	Cr—Jaffe reaction Cys C, SDMA—ELISA eGFR—Schwartz equation	elevated SDMA in patients with CKD difference in SDMA concentrations between CKD stages 1 + 2 and CKD stage 3 correlation of SDMA with creatinine in both groups and with eGFR in CKD patients SDMA was better than Cr_s_ in diagnosing CKD and in detecting CKD stages 1 and 2
	Brooks E. R. et al., 2018 [31]	352 CKD cases	ADMA, SDMA—LC–MS/MS eGFR—function of iohexol plasma clearance	inverse relation between baseline dimethylarginines levels and initial eGFR but not with rate of decline in eGFR
	Snauwaert E. et al., 2018 [32]	65 CKD stage 1–5 cases	Cr—enzymatic analysis ADMA, SDMA—ELISA eGFR—Schwartz, FAS, and β-2 microglobulin-based equation	good correlation of SDMA with eGFR poor correlation of ADMA with eGFR
	Snauwaert E. et al., 2018 [33]	57 CKD stage 1–5 cases 50 controls	Cr—enzymatic analysis ADMA, SDMA—ELISA eGFR—Schwartz equation	elevated SDMA in patients with CKD
	Benito S. et al., 2018 [34]	32 CKD stage 2–5 cases 24 controls	Cr, Cys C, ADMA, SDMA—LC/QTOF method	elevated ADMA and SDMA in patients with CKD ADMA and SDMA have similar efficacy to Cr_s_ in the early diagnosis of CKD increase in CKD classification accuracy of 18% when using 4 metabolites together (Cr_s_, SDMA, citrulline, and S-adenosylmethionine) compared to Cr_s_ alone
**CKD,** **CVD**	Brooks E. R. et al., 2009 [36]	28 CKD stage 2–3 cases 10 siblings	Cr—Jaffe reaction ADMA, SDMA—HPLC–MS/MS eGFR—Schwartz equation BP—ABPM	elevated ADMA, SDMA, and SDMA/ADMA ratio in patients with CKD correlation of SDMA, SDMA/ADMA, and L-arg/SDMA with eGFR in both groups correlation of SDMA and SDMA/ADMA with AW and AS, S, and D BP loads in both groups SDMA accounts for 27% of the variability in DBP load in AW in CKD patients with constant eGFR
	Lin I. C. et al., 2016 [37]	55 CKD stage 1–3 cases	ADMA, SDMA—HPLC BP—ABPM	correlation of ADMA with S and D BP loads
	Chien S. J. et al., 2015 [38]	57 CKD stage 1–3 cases	ADMA, SDMA—HPLC BP—ABPM	correlation of ADMA, SDMA, and L-arg/ADMA ratio with arterial stiffness correlation of L-arg/ADMA ratio with SBP and left ventricular mass
	Hsu C.N. et al., 2019 [39]	125 CKD stage 1–4 cases	ADMA, SDMA—HPLC BP—ABPM	L-arg/ADMA ratio was lower and ADMA/SDMA ratio was higher in children with abnormal office BP values
	Kuo H.C. et al., 2012 [40]	45 CKD stage 1–4 cases	ADMA_u_, SDMA_u_—HPLC BP—ABPM	Arg_u_/ADMA_u_ ratio was higher, and the ADMA_u_/SDMA_u_ ratio was lower in children with CKD stages 2–4 compared to CKD stage 1 correlation of higher (ADMA_u_ + SDMA_u_)/Arg_u_ ratio with abnormalities in ABPM
	Lin Y.J. et al., 2013 [41]	44 CKD stage 1–3 cases	Cr—Jaffe reaction eGFR—Schwartz equation ADMA, SDMA—HPLC BP—ABPM	no correlation of ADMA, SDMA, and their combined ratios with eGFR and BP loads
	Speer T. et al., 2013 [42]	22 child CKD cases 45 adult CKD cases 25 controls (15 adults, 10 children)	SDMA—HPLC–ESI–MS/MS	SDMA causes transformation from physiological HDL into “toxic” HDL that induces endothelial dysfunction
	Shroff R. et al., 2014 [43]	82 CKD stage 2–5 cases 12 controls	SDMA—HPLC–ESI–MS/MS	SDMA and SDMA in HDL elevated in patients with CKD
	Hsu C.N. et al., 2020 [44]	112 CKD stage 1–4 cases	ADMA, SDMA—HPLC eGFR—Schwartz equation	ADMA and SDMA were reference points for acrylamide metabolites in urine assay higher SDMA level in patients with CKD stages 2–4 than with CKD stage 1
	Drożdż D. et al., 2018 [45]	59 CKD stage 1–5 cases	ADMA—ELISA	ADMA was a reference point for thrombomodulin assay
	Makulska I. et al., 2015 [46]	76 CKD cases 26 controls	ADMA, SDMA—ELISA	ADMA and SDMA were reference points for sAF assay
**RT**	Andrade et al., 2011 [47]	26 RT cases 30 controls	ADMA, ADMA_u_—ELISA	elevated ADMA in patients after RT L-arg/ADMA decreased in patients after RT
**SFK**	Taranta-Janusz K. et al., 2012 [49]	51 SFK cases 21 controls	Cr—Jaffe reaction SDMA—ELISA eGFR—endogenous creatinine clearance and Schwartz equation	elevated SDMA in children with SFK no difference in SDMA concentration between URA and UN groups correlation of SDMA with creatinine and eGFR in patients with SFK
**NS**	Hyla-Klekot L. et al., 2015 [50]	32 remission and relapse NS cases	Cr, eGFR—lack of information ADMA, SDMA—HPLC	correlation of SDMA with eGFR in all patients correlation of ADMA with cholesterol, triglycerides, and traditional CVD risk factors but only in relapse
**FSGS**	Lücke T. et al., 2008 [51]	9 FSGS cases 11 other kidney disease cases 9 controls	Cr_u_—HPLC eGFR—Schwartz equation ADMA, ADMA_u_—GC–MS/MS	elevated ADMA in children with FSGS elevated ADMA_u_ excretion in children with FSGS correlation of ADMA with eGFR in children with FSGS
**HUS**	Kanzelmyere N. K. et al., 2014 [52]	12 typical HUS cases 12 controls	ADMA, ADMA_u_—GC–MS/MS	ADMA_s_ insignificantly lower in children with HUS and ADMA_u_ significantly lower in children with HUS

Abbreviations: CKD—chronic kidney disease, CVD—cardiovascular disease, RT—renal transplant, SFK—solitary functioning kidney, NS—nephrotic syndrome, FSGS—sporadic focal segmental glomerulosclerosis, HUS—haemolytic–uraemic syndrome, Cr—serum creatinine concentration, Cr_u_—urinary creatinine concentration, Cys C—serum cystatin C concentration, eGFR—estimated glomerular filtration rate, L-arg—serum L-arginine concentration, L-arg_u_—urinary L-arginine concentration, ADMA—serum asymmetric dimethylarginine concentration, ADMA_u_—urinary asymmetric dimethylarginine concentration, SDMA—serum symmetric dimethylarginine concentration, SDMA_u_—urinary symmetric dimethylarginine concentration, ELISA—enzyme-linked immunosorbent assay, HPLC—high-performance liquid chromatography, MS—mass spectrometry, LC/QTOF—an ion-pair reversed-phase liquid chromatography-quadrupole time-of-flight mass spectrometry, AW—awake, AS—asleep, S—systolic, D—diastolic, BP—blood pressure, ABPM—24 h ambulatory blood pressure monitoring, sAF—skin autofluorescence, URA—unilateral renal agenesis, UN—unilateral nephrectomy.

## Data Availability

Not applicable.

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
