# Peer review of "Asymmetric and Symmetric Dimethylarginines as Renal Function Parameters in Paediatric Kidney Diseases: A Literature Review from 2003 to 2022"

_children, 2022, doi:10.3390/children9111668_

Round 1

Reviewer 1 Report

In this manuscript the authors discussing asymmetric dimethylarginine (ADMA) and symmetric dimethylarginine (SDMA) and their physiological roles in the human body. The authors and other papers previously hypothesized that measurement of ADMA or SDMA could predict possible kidney injury. Those parameters correlates with hypertension and cardiovascular morbidity, too. The authors did a literature review.

The problem is with this manuscript, that this doesn’t contain any new information about this; furthermore, in the year, 2021 MDPI Children published quite the same paper. The weakness of this paper is that the conclusion doesn’t contain any new recommendations or new findings. I believe it would be more publishable if the authors could give more strength to this manuscript. I found only a few misspellings that need to correct.

Reviewer 2 Report

I think the organization of the this review should be organized. 

Reviewer 3 Report

Dear authors,

I want to congratulate for your outstanding work. Thank you for the privilege of doing the peer review of your paper. The manuscript is an extensive review regarding the role of dimethylarginines in CKD in children. The paper is very well written with a clear overview of the main aspects regardig this issue. 

Author Response

Dear Reviewer

Thank you for your good, positive opinion and all comments.

Round 2

Reviewer 1 Report

The authors answered the raised questions and corrected their manuscript. 

I have no more questions.